



**Global patterns and drivers of phosphorus pools in natural soils**
Xianjin He[1]; Laurent Augusto[2]; Daniel S. Goll[3]; Bruno Ringeval[2]; Ying-Ping Wang[4]; Julian Helfenstein[5,]
[6]; Yuanyuan Huang[4]; Enqing Hou[7]
1 Key Laboratory of the Three Gorges Reservoir Region's Eco-Environment, Ministry of Education, Chongqing University,

5       Chongqing, China

ISPA, Bordeaux Sciences Agro, INRAE, F-33140, Villenave d'Ornon, France
Université Paris Saclay, CEA-CNRS-UVSQ, LSCE/IPSL, Gif sur Yvette, France
CSIRO Oceans and Atmosphere, Aspendale, Vic., Australia
Agroscope, 8046 Zürich, Switzerland
Soil Geography and Landscape Group, Wageningen University, 6700 AA Wageningen, The Netherlands
Key Laboratory of Vegetation Restoration and Management of Degraded Ecosystems, South China Botanical Garden,
Chinese Academy of Sciences, Guangzhou, China
*Corresponding to*: Daniel S. Goll (*dsgoll123@gmail.com*); Enqing Hou (*houeq@scbg.ac.cn*).
**Abstract.** Most phosphorus (P) in soils is unavailable for direct biological uptake as it is locked within primary or secondary
mineral particles, adsorbed to mineral surfaces, or immobilized inside of organic material. Deciphering the composition of
different P pools in soil is critical for understanding P bioavailability and its underlying dynamics. However, widely used
global estimates of different soil P pools are based on a dataset containing few measurements in which many regions or soil
types are unrepresented. This poses a major source of uncertainty in assessments that rely on these estimates to quantify soil P
constraints on biological activity controlling global food production and terrestrial carbon balance. To address this issue, we
consolidated a database of six major soil P pools containing 1857 entries from globally distributed (semi-)natural soils and 11
related environmental variables. The P pools (labile inorganic P (Pi), labile organic P (Po), moderately labile Pi, moderately
labile Po, primary mineral P, and occluded P) were measured using a sequential P fractionation method. Using the database,
we trained random forest regression models for each of the P pools and captured observed variation with $R^2$ higher than 60%.
We identified total soil P concentration as the most important predictor of all soil P pool concentrations, except for primary
mineral P concentration, which is primarily controlled by soil pH. When expressed in relative concentrations (*i.e.,* as a
proportion of total P), the model showed that soil pH is the most important predictor for proportions of all soil P pools, except
for labile Pi proportion, which is primarily controlled by soil depth. Using the trained random forest models, we predicted soil
P pools' distributions in natural systems at a resolution of $0.5° \times 0.5°$. Our global maps of different P pools in soils as well as
the pools' underlying drivers can inform assessments of the role of natural P availability for ecosystem productivity, climate
change mitigation, and the functioning of the Earth system.






## 1 Introduction

Phosphorus (P) is a key nutrient limiting plant growth across a wide range of ecosystems (Augusto et al., 2017; Elser et al., 2007; Hou et al., 2020). Soil is typically the major P source for plants in natural terrestrial ecosystems (Weihrauch and Opp, 2018). P supplied by the soil plays a vital role in determining the structures, functions, and processes in terrestrial ecosystems (Peltzer et al., 2010; Wardle et al., 2004). For example, soil P availability imposes a major constraint on plant productivity in terrestrial ecosystems worldwide (Augusto et al., 2017; Ellsworth et al., 2022; Elser et al., 2007; Hou et al., 2020; Hou et al., 2021) and affects modeled projections of terrestrial carbon cycle responses to climate change and increasing atmospheric carbon dioxide concentrations (Cunha et al., 2022; Fleischer et al., 2019; Goll et al., 2012). The size of soil P stocks is large compared to annual plant P requirements (Wang et al., 2018) and the amount of P stored in vegetation (Wang et al., 2018; Zhang et al., 2021). However, only a small proportion of soil P can be directly taken up by plants (Morel et al., 2014), with most P tightly sorbed to soil minerals, organic compounds, or organo-mineral complexes with a turnover time of centuries to millennia or longer (Helfenstein et al., 2020; Vitousek et al., 2010). Consequently, vegetation growth is often limited by P availability in ecosystems across the globe (Vitousek et al., 2010; Wardle et al., 2004). For these reasons, the investigation of P dynamics and P bioavailability in the soil requires the identification and separation of different soil P pools (Crews et al., 1995; Walker and Syers, 1976).

Our knowledge of the various pools of P occurring in the soil is based on a limited number of chronosequence studies that investigated how P is cycled during pedogenesis (Crews et al., 1995; Walker and Syers, 1976). These studies revealed that chemical weathering results in the release of P from primary minerals, after which it can be converted to organic P through biological uptake, sorbed to soil particles, or occluded within secondary minerals. The most commonly used procedures for the sequential fractionation of P in soils were developed by Hedley et al. (1982) and later modified by Tiessen and Moir (1993). In addition to forming the basis for modeling soil P dynamics, these procedures yield operationally defined pools that are used to assess soil fertility and soil development (Wang et al., 2010; Wang et al., 2022). Pools that are commonly considered are resin extractable P; 0.5 M $NaCO_3$ extractable inorganic P (Pi) and organic P (Po); 0.1 M NaOH extractable Pi and Po; 1 M HCl-extractable P; and the occluded pool, which is composed of P remaining after extraction (see section 2.1). Several studies have called the validity of sequential extractions into question, pointing out that, while it is often assumed that pools from sequential extractions contain distinct forms of P, the reality is much more complex (Condron and Newman, 2011; Gu and Margenot, 2021; Klotzbücher et al., 2019). Nevertheless, radioisotope tracer experiments show that sequentially extracted pools have distinct P exchange behaviors that result in significantly different turnover times (Buendía et al., 2010; Bünemann et al., 2004; Helfenstein et al., 2021; Helfenstein et al., 2020; Vu et al., 2010).

Numerous studies have used data from P fractionations to explore drivers of spatial differences in soil P pools from local to global scales (*e.g.*, Brucker and Spohn, 2019; Hou et al., 2018a; Yang and Post, 2011; Chen et al., 2015). Yang and Post (2011) compiled Hedley P pools data from 178 soil samples to explore P dynamics along a soil development gradient (Yang and Post, 2011). Their results generally supported the conceptual model proposed by Walker and Syers (1976): the gradual decrease of primary mineral-bound P; the continual increase and eventual dominance of occluded P; and the overall decrease of total P as pedogenesis progresses. However, this model was challenged by Yang and Post (2011), who found that labile Pi and moderately labile Pi (non-occluded P in Walker and Syers' model) formed a significant fraction of total P at every stage of pedogenic development. Augusto et al. (2017) aggregated 1684 measurements of P pools that were taken from across the globe using the Hedley fractionation method. This work found that total P content was a main factor determining the concentrations of labile Pi and organic P pools (Augusto et al., 2017). Almost concomitantly, Hou et al. (2018) used a global dataset compiled from analyses of 802 soil samples to examine climate effects on the soil P cycle and P availability and found that soil labile Pi concentration decreased with increasing mean annual temperature and precipitation (Hou et al., 2018a). Although those studies advanced our understanding of factors controlling the size of various soil P pools, their focus was largely contained to the effects of climatic factors or soil weathering stage on a few select P pools, mainly labile Pi, and organic P. Thus, we still lack

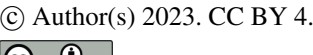



a comprehensive understanding of the relationships between environmental drivers and the various soil P pools at a global
scale.

Despite significant efforts to synthesize global Hedley soil P pool data, to our knowledge, only one set of global estimates
across natural terrestrial ecosystems exists, and this work was based on the upscaling of measurements taken from only 178
samples (Yang et al., 2013). These global estimates and associated maps have been used to explore global patterns of soil P
supply and to estimate P availability in natural and managed systems (*e.g.*, Ringeval et al., 2017; Sun et al., 2017). They have
also been used to calibrate or initialize a range of global P models (Wang et al., 2010; Yang et al., 2014). However, the poor
global coverage of the underlying data introduces significant uncertainty, potentially resulting in misinformed model
predictions and assessments.

We recently developed a new global map of soil total P concentrations and explored the underlying drivers, taking
advantage of improved data availability and the use of non-linear statistical modeling (He et al., 2021). Here, we constructed
a database of soil P pools in 1857 globally distributed (semi-)natural soils collected from 274 published studies, one order of
magnitude larger than the dataset used by Yang et al. (2013) (see comparison in Fig. S1). Using this database, we trained
random forest models to capture observed variations in Hedley P pool concentrations at the site level with two aims: (1) to
quantify the relative importance of different drivers of spatial variation in each soil P pool and (2) to develop global distribution
maps of various P pools at a spatial resolution of 0.5°×0.5° using the calibrated random forest regression model.

## 2 Material and Methods

### 2.1 Soil P fractionation terminology and procedure

The method developed by Hedley et al. (1982) and modified by Tiessen and Moir (1993) (Fig. S1) is the most commonly
used procedure for the sequential chemical fractionation of P in soils. This method exploits differences in solubility to separate
different 'forms' of P occurring in the soil. Though it cannot be used to identify specific discrete P compounds in the soil, this
approach has proven indispensable for the study of soil P cycling and, as such, is widely used (Condron and Newman, 2011;
Klotzbücher et al., 2019; Barrow et al., 2021). Here, we use the word 'pool' to indicate the concentrations quantified in each
step during sequential fractionation. We use the word 'proportion' to represent the size of a pool relative to total P.

There is disagreement about how to interpret the different pools yielded by sequential fractionation (Gu et al. 2019;
Barrow et al., 2021; Klotzbücher et al., 2019; Condron and Newman, 2011; Helfenstein et al., 2020). Here, we adopt a widely
used regime for understanding these pools: The resin Pi pool represents the soil soluble Pi pool, which is immediately
accessibly to plants; The $HCO_3^-$ Pi pool can be released by ligand exchange with bicarbonate ions; This pool is available to
plants and persists for only short periods (*e.g.*, a growing season); Due to their functional similarity, the resin and $HCO_3^-$ Pi
pools can be combined and used as an index of labile inorganic P (*i.e.*, 'available' P); The $HCO_3^-$ Po pool represents labile Po
that can be utilized by plants after mineralization. The $OH^-$ P (Pi and Po) pools indicate moderately labile P that is bound to
both amorphous and crystalline Al and Fe. This pool represents P that is moderately available to plants; The 1 M HCl Pi pool
represents primary mineral P that is bound to calcium and that can be utilized by plants after it is released by weathering; And
other P pools, such as residual P, are least available to plants due to their particularly low solubility.

To integrate data from studies that use different interpretations, we consider a set of six simplified P pools (Fig. S3):
labile Pi, labile Po, moderately labile Pi, moderately labile Po, primary mineral P, and occluded P. Labile Pi includes the resin
Pi and $HCO_3^-$ Pi pools; labile Po and moderately labile Po are organic pools extracted by carbonate and NaOH, respectively;
moderately labile Pi is the NaOH Pi fraction; primary mineral P represents the 1 M HCl Pi pool; and occluded P includes any
remaining P (Hou et al., 2018b).

We collected, filtered, and processed soil P pool data (see section 2.2.) from the literature (Supplementary material).



First, we added all measured P pools together to calculate total soil P, unless any pools were missing. In this case, we instead
used the measured value of total soil P presented in that paper. Second, if phosphate was extracted using deionized water before
the resin P extraction step, the labile Pi pool includes both resin and aqueous P. If the extraction procedure began by using
sodium bicarbonate solution instead of resin P, we classified $HCO_3^-$ Pi as labile Pi. Third, the labile Po pool and the moderately
labile Po pool represent the $HCO_3^-$ Po and NaOH pools, respectively. The raw data contained other organic P pools (*e.g.,* Po
extracted by sonication and NaOH or by hot, concentrated HCl) which we included as part of occluded P. Fourth, if occluded
P was not reported, we calculated this pool's concentration by subtracting the sum of the five other pools from total P.
**2.2 Data source and processing**
We collected soil P pool data by aggregating all the publications that cited either one of two main references dedicated
to Hedley's method (Hedley et al., 1982; Tiessen and Moir, 1993). We included all studies that reported data from (semi-)
natural soils that supported primary vegetation or that had been reforested with a stand older than 10 years and no documented
history of P fertilization. We excluded observations taken from pot experiments, mine zones, and intertidal zones, as P pools
in these soils could be affected by factors different from those influencing (semi-) natural soils. Despite our best efforts, we
cannot rule out that our database includes data collected from soils affected by undocumented anthropogenic activities in the
past (*e.g.*, P fertilization occurring before reforestation), particularly in western Europe and eastern USA (*e.g.*, De Schrijver et
al., 2012). All data were collected at the plot scale. For data that included replicates within a plot or soil layer, average values
were calculated.
To compile our database, we first combined the two existing global databases (Augusto et al., 2017; Hou et al., 2018b).
Detailed information about the methods used to construct these datasets can be found in the original publications. We extracted
observations from these two databases by selecting only unfertilized, uncultivated, and (semi-) natural soils. This yielded 1684
observations from 182 studies from the dataset developed by Augusto et al. (2017) and 802 observations from 99 studies from
the dataset developed by Hou et al. (2018). Next, we removed 375 duplicates, after which our dataset contained 2111
observations from 245 studies (Figure S2). Because we use total soil P concentration as a predictor of soil P pools, we removed
data that did not include total soil P (calculated as the sum of P pools or measured by a separate method) or that did not identify
the concentration of at least one pool (*e.g.*, labile Pi, labile Po, moderately labile Pi, moderately labile Po, primary mineral P,
or occluded P). In this step, 816 observations were removed, resulting in a dataset that included 1295 observations from 178
studies.
Next, we added additional observations by compiling data from literature published after 2016, the final year included
in the database compiled by Hou et al. (2018). We used Google Scholar to search for studies published between 2016 and
08/08/2021 that referenced either Hedley et al. (1982) or Tiessen & Moir (1993). This search returned 701 publications citing
Hedley et al. (1982) and 245 citing Tiessen & Moir (1993). From this set, we selected studies that presented soil P data collected
using the fractionation method for (semi-)natural soils. The resulting 562 observations from 96 studies were added to our final
dataset, which includes a total of 1857 observations collected from 729 sites from 274 studies (Supplementary Text 1 and Fig.
S1).
In addition to soil P pool concentration and site coordinates, our dataset contains site characteristics including climate
variables (*i.e.*, mean annual temperature (MAT), mean annual precipitation (MAP), and potential biome), soil physiochemical
properties (*e.g.*, soil organic carbon concentration (SOC), soil clay and sand content, and soil pH), and elevation (Table 1).
Potential biome was identified using a global map of potential natural biomes (*i.e.*, the global distribution of biomes that would
exist in the absence of human activity) (Hengl et al., 2018). This categorization includes seven ecosystem types, including
tropical forest, temperate forest, boreal forest, grassland, savanna, desert, and tundra. We did not include parent material type
because it can be inferred from soil total P concentration and other soil properties (*e.g.*, soil texture and pH) (Augusto et al.,
2017; He et al., 2021). Because soil age was rarely reported, we used USDA soil order identity as a proxy for 3 age classes:



slightly, intermediately, and strongly weathered (Smeck, 1985; Yang et al., 2013). Among the 12 USDA soil orders, Entisols,
Inceptisols, Histosols, Andisols, and Gelisols are classified as slightly weathered soils. Alfisols, Mollisols, Aridisols, and
Vertisols are classified as intermediately weathered soils. Oxisols, Ultisols, and Spodosols are classified as strongly weathered
soils (Yang et al., 2013; Smeck, 1985). Given that atmospheric P inputs are small (0.3 kg P ha yr$^{-1}$, on average) compared to
soil P stocks (Mahowald et al., 2008; Wang et al., 2015) and are also highly uncertain over timescales relevant to soil
development, we do not consider atmospheric inputs as a predictor of P pools. As such, we did not include this information in
our dataset. We extracted data from each publication as available. In cases in which relevant information was not reported, we
extracted the missing data from gridded datasets (Table S1) based on the geographic coordinates of the study sites.

In random forest modelling, correlated predictors can be substituted for each other so that the importance of correlated
predictors will be shared, making each predictor's estimated importance smaller than its true value (Strobl et al., 2008). Thus,
we did not include soil total nitrogen content as it is strongly correlated with SOC ($r$ = +0.94), nor did we include aridity index
as it is strongly correlated with MAP ($r$ = +0.72). We also did not include rarely reported variables that were included in the
referenced studies (*e.g.*, soil extractable aluminum and iron concentrations).
**2.3 Statistical modelling**

All statistical analyses and plotting were performed in the R environment (v. 4.0.2) (R Core Team, 2018).
The database includes some extreme values in each P pool (Table 2). These values were likely observed in exceptional
geological contexts (Porder and Ramachandran, 2013) or in special soils (*e.g.*, very young volcanic soils). We included these
extreme values in the shared version of the dataset and in the database summary (Table 2). However, these values were
excluded from data used in model training, as the extremely high values could have a large influence on modeled relationships
between soil P pools and predictors. To this end, we only included values falling in the interval between 1% and 99%.
We used random forest regression models (Breiman, 2001) to predict global patterns of distribution for individual soil P
pools. All models included the same 11 predictors: MAT, MAP, potential biome, total P, soil depth, SOC, soil clay and sand
content, soil pH, elevation, and soil weathering stage (Table S1). The random forest analysis accounts for interactions and
nonlinear relationships between predictors and is appropriate for handling the multicollinearity problem in the multivariate
regression (Delgado-Baquerizo et al., 2017). The fit for each tree was determined by randomly selecting test cases. In order to
compare the relative importance of different variables for predicting the size of a pool, we performed random forest regression
analysis using the R package *caret* by applying the embedded R package *randomForest* version 3.1 (Liaw and Wiener, 2002)
with an automated *mtry* parameter. Five-fold cross-validation was performed using the R package *caret* (v. 6.0-86) (Kuhn,
2020) to evaluate model performance. The mean decrease in accuracy (%IncMSE) was used to evaluate the relative importance
of each variable as a predictor of a soil P pool. The mean decrease in accuracy plot shows how the accuracy of the fitted model
declines with the exclusion of a predictor. The greater the decline in accuracy, the more important the variable is for prediction.
In this study, the importance measure was calculated for each tree and averaged across the forest (500 trees). Our model found
that all 11 variables are important for predicting pool concentrations; thus, all were used as predictors as we developed the
global distribution map. Partial dependence plots showed the marginal effect of each continuous predictor on soil P pool
concentration. We used the *partial_dependence* function in the R package *edarf* version 1.1.1 (Jones and Linder, 2016) to
calculate the partial dependence of the response on an arbitrary dimensional set of continuous predictors from a fitted random
forest model.
Finally, we used the built forest regression models for each of the soil P pools for global prediction by using the predict
function in the *ranger* package with globally gridded datasets for all 11 variables. The *predict* function in the *ranger* package
(Wright and Ziegler, 2017) can compute the standard error of a predicted value. To estimate standard errors based on out-of-
bag predictions, we used the infinitesimal jacknife for bagging approach (Wager et al., 2014). Prior to use for global predictions,





driver variables were re-gridded to the same spatial resolution of 0.5° × 0.5° (the original resolution of each predictor can be
found in Table S1). We did not mask croplands or other areas heavily influenced by human activity (*e.g.*, urban areas), so pool
concentrations predicted for these regions should be interpreted as the natural state prior to anthropogenic activity.

Because we trained models to predict P pool concentrations and proportions using the same 11 variables, we had two

options for developing global maps of P pool proportions: (1) dividing a pool's concentration by total P (He et al., 2021), or
(2) using our trained model. The resulting maps (Fig. S6) are highly correlated, with Pearson correlation coefficients from 0.61
to 0.98. Model accuracy was higher for predicted concentrations than it was for predicted proportions. (Fig. 2 & 3). Therefore,
we developed our map using the model to predict P pool concentration, after which these predictions were used with total soil
P concentrations to calculate P pool proportions (He et al., 2021) rather than predicting them using random forest models.

Soil depth was used as a predictor, allowing models to predict soil P pool concentration for any given depth (Hengl et al.,

2017). The partial dependence plot indicated that soil P pool concentration decreased approximately linearly with soil depth
in the top 30 cm and that there was no apparent trend with depth in the subsoil (30-100 cm) (Fig. S2-7); As such, we calculated
global soil P pool concentration at 0 cm, 10 cm, 20 cm, 30cm, and 100 cm. Averages for a depth interval (*e.g.*, 0-30 cm or 0-
100 cm) can be derived by calculating the weighted average of the predictions within that interval (Hengl et al., 2017).
**3 Results**
**3.1 Characters of P pools in natural soils across the world**

Our soil P pool database includes 1857 measurements from 729 geographically distinct sites and covers 6 continents, all

major biomes, and all 12 USDA soil orders in terrestrial ecosystems (Fig. 1). The database includes pool concentrations
measured in samples collected from the topsoil (0-30 cm) to a depth of 450 cm, with 83% of the measurements taken from the
topsoil. In the database, both concentrations and proportions of all P pools were highly skewed to the right (Fig. S4 & S5).

The largest pool among the six pools considered is occluded P, with a global mean concentration of 260.5 mg kg$^{-1}$ and a

mean proportion of 41.9% (Table 2). Primary mineral P has a global mean concentration of 106.8 mg kg$^{-1}$ and a mean proportion
of 19.0%. The labile and moderately labile Po pools have global mean concentrations of 31.1 mg kg$^{-1}$and 120.3 mg
kg$^{-1}$, respectively, and mean proportions of 5.9% and 19.5%, respectively. Moderately labile Pi has a global mean concentration
of 58.7 mg kg$^{-1}$ and a mean proportion of 9.3%. Labile Pi represents the smallest proportion of total P with a global mean
concentration of 37.1 mg kg$^{-1}$ and a mean proportion of 5.9% (Table 2).
**3.2 Model performance of different P pools in soils**

The random forest regression models explained 62%, 64%, 60%, 83%, 76%, and 82% of the variance in the concentrations

of labile Pi, labile Po, moderately labile Pi, moderately labile Po, primary P, and occluded P, respectively (Fig. 2). Using the
importance measure (%IncMSE), we identified the most important predictor for concentrations of soil labile Pi, labile Po,
moderately labile Pi, moderately labile Po, and occluded P as total P concentration, while the most important predictor for soil
primary P was the soil pH (Fig. 2). The random forest regression models explained 48%, 58%, 52%, 64%, 80%, and 58% of
the variance in proportions of labile Pi, labile Po, moderately labile Pi, moderately labile Po, primary P, and occluded P,
respectively (Fig. 3). Based on the importance measure, the most important predictor for proportions of soil labile Po,
moderately labile Pi, moderately labile Po, primary P, and occluded P was soil pH, while the most important predictor for labile
Pi proportion was soil depth (Fig. 3).
**3.3 Global patterns of P pools in natural soils**



The maps we developed indicate that soil P pool concentrations showed substantial differences between biomes (Fig. 4).

Lower P pools concentrations were found in warm and/or humid biomes (*e.g.*, tropical forest and savanna), while higher P

pool concentrations were found in northern cold biomes (*e.g.*, tundra and boreal forest) (Fig. 4C). Estimated subsoil P pool

concentrations showed similar patterns to those identified in the topsoil. The spatial patterns of pool proportions were different

from those of pool concentrations across biomes (Fig. 7 & 8). For example, variation in the proportion of P held in the labile

Pi pool was relatively small compared to variation observed in pool concentrations; moreover, the proportion of occluded P

tended to increase in the transition from tundra and boreal forest to tropical forest and savanna (Fig. 4D). It should be noted

that the mapped predictions of P pool concentrations across biomes are not consistent with the measured data (Fig. S4), which

indicate that total soil P in tropical forests is higher than in any other biome. This result suggests a sampling bias due to

overrepresentation of high total soil P sites in the tropical forest data.

Our global predictions revealed that average values across all P pools were higher in slightly weathered soils compared

to those in strongly weathered soils (Fig. 4A), reflecting the strong effect of soil development on soil P depletion. While

occluded P proportion increased with soil development, the proportions of labile and moderately labile P (Pi and Po) were

fairly independent of soil weathering stage (Fig. 4B).

There are significant differences between our predictions and those made by Yang et al. (2013) in both the magnitude and

the spatial patterns associated with most P pool concentrations. The two global estimates were only weakly to moderately

correlated (Pearson correlation coefficients between 0.09 and 0.38) (Fig. S8). Yang et al.'s predictions are lower than ours for

organic P, moderately labile Pi, primary mineral P, and occluded P concentrations (Table S5). Although average values for

labile Pi concentrations estimated by Yang *et al.* were close to ours, they were only weakly correlated with each other (Pearson

correlation coefficient of 0.09).

Partial dependence plots (Fig. 5) and the results of Pearson correlation analysis (Table 3) revealed that total P

concentration was significantly and positively correlated with concentrations for all six pools. SOC was significantly and

negatively correlated with primary mineral P concentration, but positively correlated with the other five pool concentrations.

MAT and MAP were significantly and negatively correlated with concentrations of all soil P pools. Soil pH was significantly

and positively correlated with primary mineral P concentration, but significantly and negatively correlated with concentrations

of the other five P pools. The results of Pearson correlation analysis also indicated that P pool concentrations were well

correlated with each other, with the exception of primary mineral P; this pool was negatively correlated with labile Po and not

correlated with moderately labile Po concentration.

Partial dependence plots (Fig. 6) and Pearson correlation analysis (Table 3) revealed that soil pH was positively correlated

with the primary mineral P proportion and negatively correlated with the other five P pool proportions. Soil labile Po,

moderately labile Pi, and moderately labile Po proportions decreased substantially with an increase in MAT, while the occluded

P proportion increased with MAT. Soil labile Po, moderately labile Pi, and moderately labile Po proportions increased

substantially with increasing total P concentration, while the soil labile Pi and occluded P proportions decreased substantially

with total P concentration.

**4 Discussion**

**4.1 Improved mapping of different P pools in global natural soils**

We trained random forest regression models using 11 variables to predict six soil P pools at different depths in

(semi-)natural terrestrial ecosystems, resulting in significant improvements over earlier estimates (Yang et al., 2013). First, we

used a new global map of total P concentrations in natural soils (He et al., 2021) as a predictor. Because total P is an important



predictor and is highly correlated with all other P pools (Fig. 2 & 7), a higher quality map of total soil P provides much
improvement to the global estimates of P pool concentrations in soils (He et al., 2021). Second, Yang *et al*. (2013) used a
limited number (n=178) of measurements of Hedley P pools across soils. Our database represents a nearly ten-fold increase,
which can better represent the heterogeneous conditions on Earth. Third, Yang *et al.* (2013) estimated P pool concentrations
using total soil P concentrations, global soil order maps, and average proportions of various P pools for different soil orders.
In our study, we found that soil orders were less informative than other environmental predictors. By including more predictors
(*e.g.,* SOC, climate, and soil pH) our model offers significant improvements for capturing the variation observed in soil P
composition across the globe.
Differences between our estimates of different P pools and those presented by Yang *et al*. (2013) have significant
implications for soil P availability to vegetation. The averages and median values of Yang *et al.*'s predicted soil organic P,
moderately labile Pi, and occluded P concentrations were substantially lower than our estimates. Evidence suggests that soil
organic P and moderately labile Pi remain bioavailable on timescales of days to months (Helfenstein et al., 2020; Augusto et
al., 2017; Maharjan et al., 2018), while occluded P is bioavailable on the order of years to millennia (Hou et al., 2016; Wang
et al., 2007). Thus, soil P availability might be larger than previously assumed in assessments based on estimates by Yang *et
al.* (2013) (e.g., Sun et al. 2017).

### 4.2 Major drivers of different P pools in natural soils

Our results indicate that global variation in soil P pools is jointly controlled by total P concentration, soil pH, soil
development, climatic factors, and soil depth. Given that our models explain > 48% of the variance observed in P pools, our
results (see Figures 2 and 3) suggest that edaphic properties and climatic factors play significant roles in the size and
composition of different soil P pools globally. We discuss the effects of three major factors in more detail below.

**Effects of total soil P concentration on P pools**

We found that total soil P concentration was a prominent predictor of most soil P pools at the global scale and that total
P was positively correlated with all P pool concentrations and Po pool proportions. This is consistent with findings at local
(Turner and Blackwell, 2013) and global (Augusto et al., 2017; Hou et al., 2018; Harrison, 1987) scales. Total soil P is
influenced by multiple soil forming factors (*e.g.,* parent material P concentration, climate, soil organic carbon content, and soil
texture) (He et al., 2021). Thus, total soil P provides an integrated measure of factors that regulate the size of the P pools.
Moreover, this result is consistent with the emerging idea of substrate-based P cycling in natural ecosystems (Lang et al., 2017;
Lang et al., 2016): Soils with high total P content are usually also associated with a large primary mineral P pool. At these P-
rich sites, plant and microbial communities tend to promote P release from primary minerals, with subsequent biological and
abiotic transformations resulting in high concentrations in all other P pools (Lang et al., 2016; He et al., 2021) and higher
proportions of organic P (Hou et al., 2018c). In contrast, at P-poor sites, plant and microbial communities are more reliant on
P recycling systems that promote the mineralization of Po by soil microbes (Achat et al., 2009; Marklein and Houlton, 2012)
and the mobilization of moderately labile Pi or even occluded P (Augusto et al., 2017) to sustain the P supply. Therefore, soil
P pool concentrations are expected to co-vary with total soil P concentration.
Interestingly, our predictions indicate that labile Pi concentrations are not primarily controlled by soil P supply (*i.e.,* total
soil P) on a global scale, but by biological processes such as plant uptake, microbial uptake, immobilization, and mineralization
(Yang et al., 2013; Cross and Schlesinger, 1995; Weihrauch and Opp, 2018; Hou et al., 2016). Some work suggests that in
strongly weathered soils (with limited P stocks), mineralization of Po could be a major source of labile Pi (Vitousek, 1984;
Achat et al., 2009). In addition, fluctuating redox conditions in highly weathered soils can cause the release of labile Pi from
the moderately labile P pool through the reduction of $Fe^{3+}$ minerals (Chacon et al., 2006; Liptzin and Silver, 2009). This is
consistent with other studies that found that the labile Pi pool is quite stable across weathering stages (Cross and Schlesinger,



1995; Yang and Post, 2011; Zhang et al., 2005).

### Effects of soil pH on P pools

Consistent with previous studies (Hou et al., 2018c; Kruse et al., 2015; Oburger et al., 2011; Barrow et al., 2020), our
results indicate that soil pH is an important predictor of P pool concentrations and proportions in natural soils globally. The
relative importance of pH is unsurprising, since the sequential fractionation procedure is based on dissolving a soil sample in
solutions of varying acidity/alkalinity. However, the observed pH effects also support the existing mechanistic understanding
of the various pools. The strong positive correlation of primary P and soil pH is expected because 1) the primary P pool is
composed mainly of calcium phosphate/apatite, which is highly soluble at low pH but becomes less soluble with increasing
pH and 2) soil pH declines with soil weathering intensity (Delgado-Baquerizo et al., 2020) (*e.g.*, the highest values of soil pH
are usually found in dry regions where chemical weathering rates are limited by water availability (Slessarev et al., 2016)).
Both factors affect the transformation of primary mineral P to other forms.
Soil pH shows important but negative influences on the proportions of other soil P pools (*i.e.*, proportions of labile Pi and
Po, moderately labile Pi and Po, and occluded P). There are several possible explanations for these relationships. First, low
soil pH (< 5.0) inhibits soil microbial activities and the extracellular activity of phosphatase enzymes (Aciego Pietri and
Brookes, 2008; Eivazi and Tabatabai, 1977; Xu et al., 2017). Thus, in acidic soils, more organic P (*i.e.*, labile Po) may
accumulate than in neutral soils. Second, decreasing soil pH is associated with the accumulation of Fe and Al oxides, which
leads to enhanced adsorption of P (*i.e.*, labile Pi and labile Po). Third, pH tends to decrease as soil weathering advances and
base cations are progressively washed out (Slessarev et al. 2016). As soils weather, occluded P accumulates, which explains
why the occluded pool decreases with increasing pH. Fourth, increasing soil pH is associated with enhanced adsorption of
dissolved Pi to Ca and Mg, reducing the amount of labile Pi available for plants and soil microorganisms (Fink et al., 2016;
Gerke, 2015). This could explain the negative relationship between soil pH and the labile Pi proportion identified in this study.

### Effects of soil development on P pools

The variation of P pools across weathering stages predicted by our model partially supports Walker and Syers' (1976)
theory based on soil chronosequences, and is consistent with more nuanced models of soil P evolution that consider variation
in tectonic uplift and geological composition (Buendía et al., 2010). While our results are consistent with expectations from
Walker and Syers' theory about the increase in the proportion of occluded P that occurs at the expense of primary and organic
P during soil development, they do not support Walker and Syers' ideas regarding the evolution of the labile Pi and moderately
labile Pi pools. The evolution of occluded P is commonly explained by the increase of Al and Fe oxide minerals and the
decrease of soil pH; In addition to being fixed onto Fe and Al oxides, P that is released from primary minerals or mineralized
from organic matter can be occluded by adsorbing to mineral surfaces (Crews et al., 1995; Selmants and Hart, 2010). This
mechanistic understanding is in line with our findings about the dependence of soil P pools on soil pH. However, our results
disagree with the prediction by Walker and Syers' (1976) model in that labile Pi and moderately labile Pi (non-occluded P in
Walker and Syers' model) formed significant fractions of total P throughout all soil orders across weathering stages. This could
be due to the coarse classification of weathering stages in our study, which may be insufficient to characterize the end members
of the range. This explanation is supported by the small proportion of 1 M HCl P in the slightly weathered soil and the moderate
amounts of P remaining in strongly weathered soils. The inconsistency may also be due to the fact that Walker and Syers'
model was based on a system with negligible tectonic uplift, as argued by Buendía et al. (2010). In Buendía *et al.*'s model, the
primary P pool is replenished by the uplift effect, a prediction that is supported by our finding of considerable primary mineral
P pools occurring across soil weathering stages.

### Effects of soil depth on P pools



We found that soil P pools varied significantly with soil depth. Total soil P in topsoil was higher than in subsoil due to biological uplift, which was reported by previous studies (Jobbágy and Jackson, 2001; Porder and Chadwick, 2009). The labile and moderately labile P pool concentrations (in both inorganic and organic pools) showed a similar trend. In contrast, the primary P and occluded P concentrations in topsoil were lower than in the subsoil. This can be explained by the fact that topsoil tends to be more weathered and developed than the subsoil (Achat et al., 2012; Chen et al., 2021).

### 4.3 Limitations and prediction uncertainty

In our database, some regions were underrepresented (*e.g.*, northern Canada, middle and northern Asia, and inner Africa), which may result in low accuracy of the predicted values in those regions. In the tropics, high P soils were overrepresented and accuracy of predicted values in tropical regions may be quite low. Our database contains four times as many observations from surface mineral soils (0-30cm) than it does from soils deeper than 30 cm. As such, the predicted concentrations of different P pools for deep soils may suffer from larger uncertainties. Finally, large portions of variation remain unexplained, especially variation in soil labile Pi concentrations and proportions (40% and 52% unexplained, respectively), indicating that other factors that were not accounted for play a role. These factors may include microbial processes, Fe and Al oxide concentrations, plant community composition, atmospheric deposition, and soil erosion (Kruse et al., 2015; Achat et al., 2016). These limitations highlight the need for additional measurements, particularly from underrepresented regions and the subsoil as well as measurements of closely associated variables, especially those related to labile Pi.

### 5 Conclusion

Here, we compiled the largest database to date of different soil P pools. Using machine learning modelling, we quantified the relative importance of multiple predictors for estimating different soil P pools and estimated these pools at the global scale. Our results indicated that the global concentrations of soil labile Pi, labile Po, moderately labile Pi, moderately labile Po, and occluded P could be predicted mainly by the total soil P concentration, while primary P concentration was mainly predicted by soil pH and total soil P concentration. For predicting proportions of total P, soil pH was the most important predictor for all P pools at the global scale, with the exception of labile Pi proportions, for which soil depth was the main driver. In addition, our results also revealed significant effects of climate and other edaphic factors on spatial variation in P pools. We concluded that edaphic properties and climatic factors were significant predictors of soil P pools, including concentration and proportion of total P. These findings represent a significant step towards improving understanding of global variations in different soil P pools. Our global maps of predictions of different P pools will be important to efforts to improve global scale biogeochemical models of the P cycle.

**Data availability**

Raw datasets, data source reference, R script, and global maps generated in this study are available at https://doi.org/10.6084/m9.figshare.16988029 (He et al., 2022).

**Author contributions**

D.S.G., Y.W. and E.H designed this study. L.A., E.H, and X.H. collected the data. X.H., E.H., L.A., D.S.G., B.R., Y.W., J.H., and Y.H. discussed analyzing methods. X.H. conducted the analysis and drafted the manuscript. All authors discussed the results and contributed to the manuscript.

**Financial support**



This research was funded by the China Postdoctoral Science Foundation (2020M673123), the National Natural Science
Foundation of China (32271644), and the ANR CLAND Convergence Institute. We would like to thank Dr. Joseph Elliot at
the University of Kansas for his assistance with English language and grammatical editing of the manuscript.
**Acknowledgements**
We would like to thank Dr. Joseph Elliot at the University of Kansas for his assistance with English language and grammatical
editing of the manuscript.
**Competing interests**
The authors declare that they have no conflict of interest.

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





**Table 1. Summary of training data used to predict soil P pool concentrations.** P10 and P90 indicate percentile rank of 10%
and 90%, respectively. Proportions from literature (PFL) and proportions from gridded maps (PFGM) indicate proportions of
measurements from the literature and extracted from global gridded maps, respectively.

| Group | Variables | Unit | Min | P10 | Median | P90 | Max | PFL[*] | PFGM[#] |
|---|---|---|---|---|---|---|---|---|---|
| Climate | MAT | °C | -12 | 1.1 | 12.8 | 25.7 | 30.0 | 96% | 4% |
| | MAP | mm yr$^{-1}$ | 10 | 414 | 970 | 2750 | 5180 | 96% | 4% |
| Soil property | Total P | mg kg$^{-1}$ | 4.8 | 114.0 | 455.5 | 1107.9 | 14973.6 | 100% | 0% |
| | SOC | g kg$^{-1}$ | <0.1 | 4.8 | 24.4 | 130 | 545.2 | 87% | 13% |
| | Soil pH | unitless | 3.0 | 4.2 | 5.7 | 8.1 | 10.5 | 92% | 8% |
| | Soil clay | g kg$^{-1}$ | <0.1 | 70.0 | 195.5 | 410.7 | 945.5 | 52% | 48% |
| | Soil sand | g kg$^{-1}$ | <0.1 | 164.9 | 420.0 | 757.6 | 982.0 | 49% | 51% |
| | Depth | cm | 0.5 | 4.2 | 10.0 | 47.5 | 450.0 | 100% | 0% |
| | Soil order | unitless | 12 USDA soil orders | | | | | 80% | 20% |
| Vegetation | Biome | unitless | 8 major biomes | | | | | 0% | 100% |
| Topography | Elevation | m | -2 | 37 | 616 | 3015 | 4813 | 85% | 15% |

MAT: Mean annual temperature; MAP: Mean annual precipitation; SOC: Soil organic carbon.





**Table 2. Statistical summary of P pools in global (semi-)natural soils.** Results based on our collected sites database. P10,
P25, P75, and P90 indicate percentile rank of 10%, 25%, 75%, and 90%, respectively.

| | Count | Min | P10 | P25 | Median | Mean | P75 | P90 | Max |
|---|---|---|---|---|---|---|---|---|---|
| *Concentration (mg kg⁻¹)* | | | | | | | | | |
| Labile Pi | 1722 | <0.1 | 2.2 | 6.2 | 14.3 | 37.1 | 34.3 | 78.6 | 961.5 |
| Labile Po | 1567 | 0.1 | 2.5 | 5.9 | 14.0 | 31.1 | 35.0 | 85.2 | 422.0 |
| Moderately labile Pi | 1742 | <0.1 | 4.0 | 10.0 | 25.0 | 58.4 | 57.7 | 122.4 | 4520.9 |
| Moderately labile Po | 1588 | 0.2 | 8.3 | 22.1 | 60.8 | 120.3 | 155.1 | 333.4 | 1876.7 |
| Primary P | 1629 | <0.1 | 1.2 | 4.7 | 38.9 | 106.8 | 145.0 | 328.3 | 1560.0 |
| Occluded P | 1453 | 0.8 | 34.5 | 86.2 | 178.0 | 260.5 | 309.6 | 532.9 | 2845.4 |
| *Proportion of total P (%)* | | | | | | | | | |
| Labile Pi | 1448 | <0.1 | 0.6 | 1.7 | 4.0 | 5.9 | 7.7 | 13.6 | 54.5 |
| Labile Po | 1331 | <0.1 | 0.8 | 1.7 | 4.1 | 5.9 | 7.8 | 13.1 | 63.4 |
| Moderately labile Pi | 1448 | <0.1 | 0.9 | 3.0 | 7.5 | 9.3 | 12.9 | 20.2 | 55.6 |
| Moderately labile Po | 1384 | 0.1 | 3.1 | 8.0 | 18.0 | 19.5 | 27.1 | 38.5 | 74.1 |
| Primary P | 1448 | <0.1 | 0.5 | 1.6 | 7.9 | 19.0 | 29.4 | 60.9 | 95.1 |
| Occluded P | 1448 | 0.8 | 15.4 | 26.8 | 42.4 | 41.9 | 56.4 | 67.9 | 92.3 |




**Table 3. Coefficients of Pearson correlations among proportions and concentrations of soil P pools.** Results based on the
predicted maps for soils at depths of 0-30 cm. Coefficients with $P < 0.001$ are shown in black and bold. Labile Pi P. indicated
the labile Pi proportion. The same meanings to the Labile Po P., Moderately labile Pi P., Moderately labile Po P., Primary P P.,
and Occluded P P..

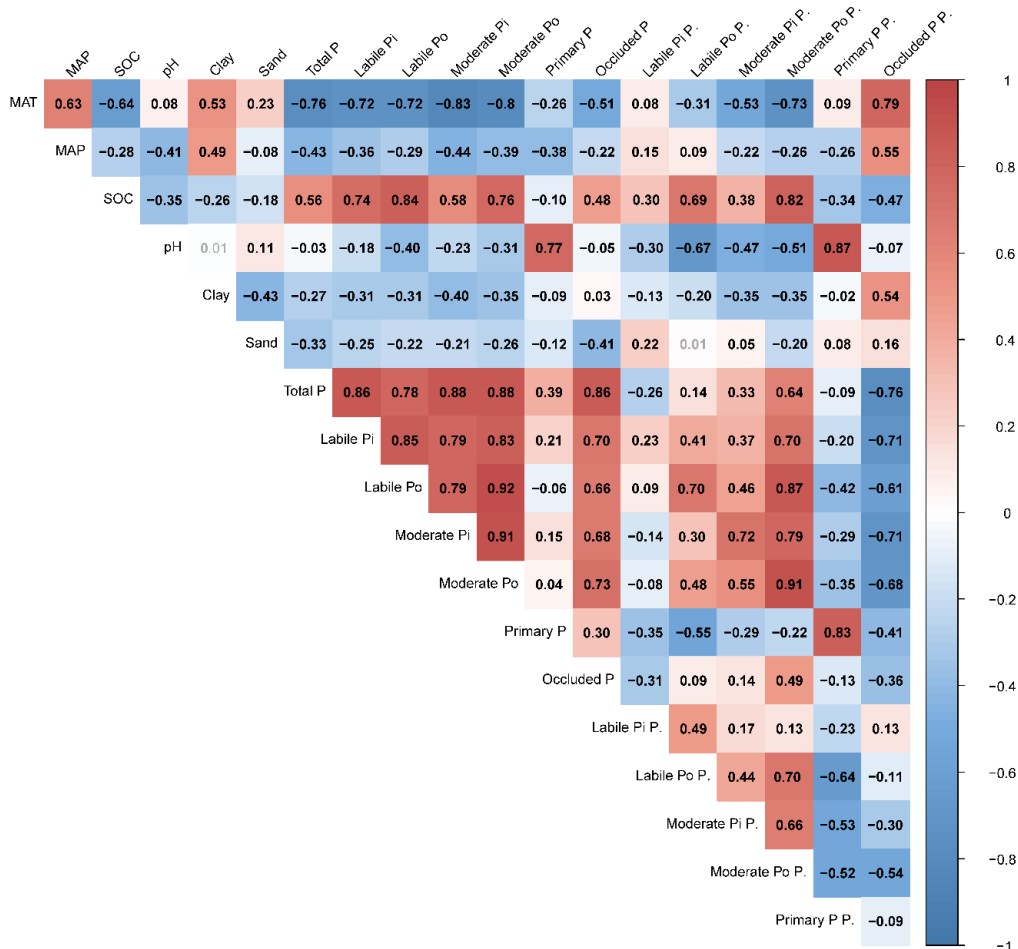




**Figure 1. Distribution of site-level training data.** The database contains 1838 observations covering 12 USDA soil orders
(B) and all major terrestrial biomes (C).

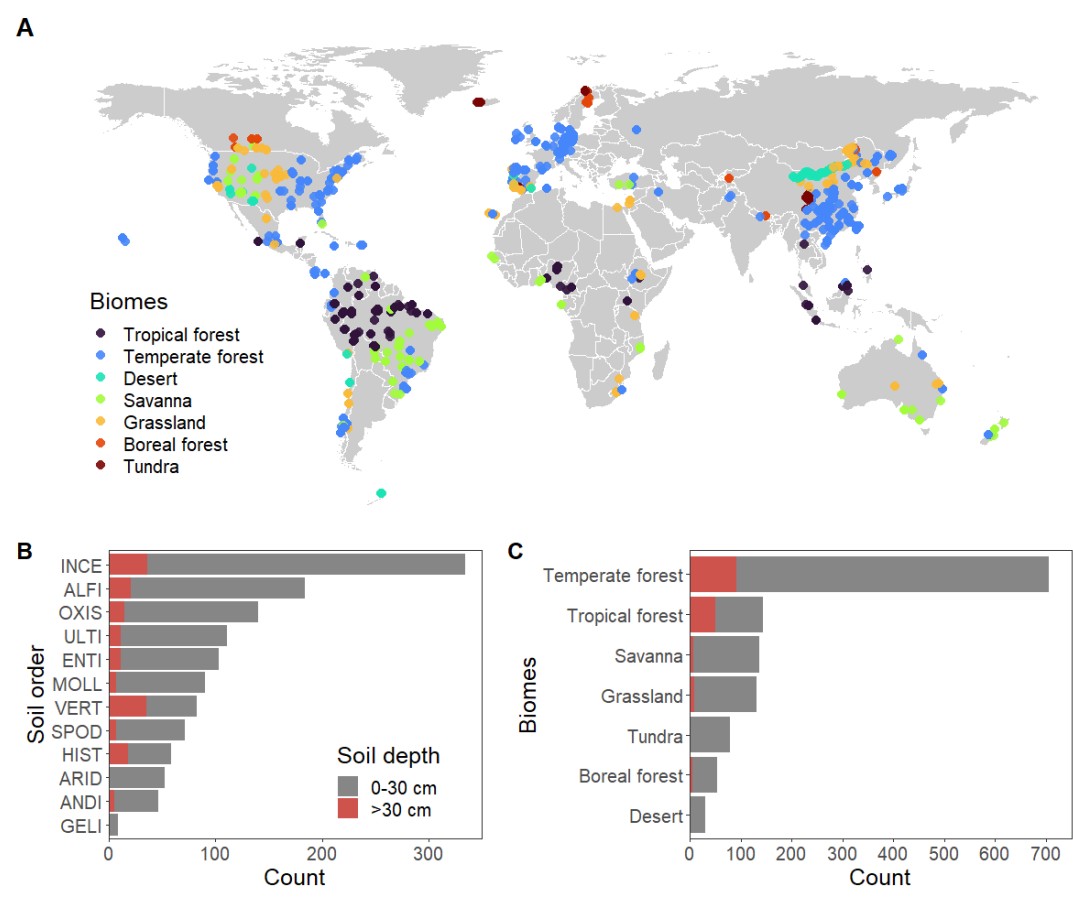






**Figure 2. Relative importance of variables for predicting concentration of soil P pools quantified using random forest**
**models.** Mean decrease accuracy (%IncMSE) indicates the relative importance of each variable for predicting soil P pools.
SWS: soil weathering stage.







**Figure 3. Relative importance of variables for predicting proportions of soil P pools quantified using random forest**
**models.** Mean decrease accuracy (%IncMSE) indicates the relative importance of each variable for predicting soil P pools.
SWS: soil weathering stage.

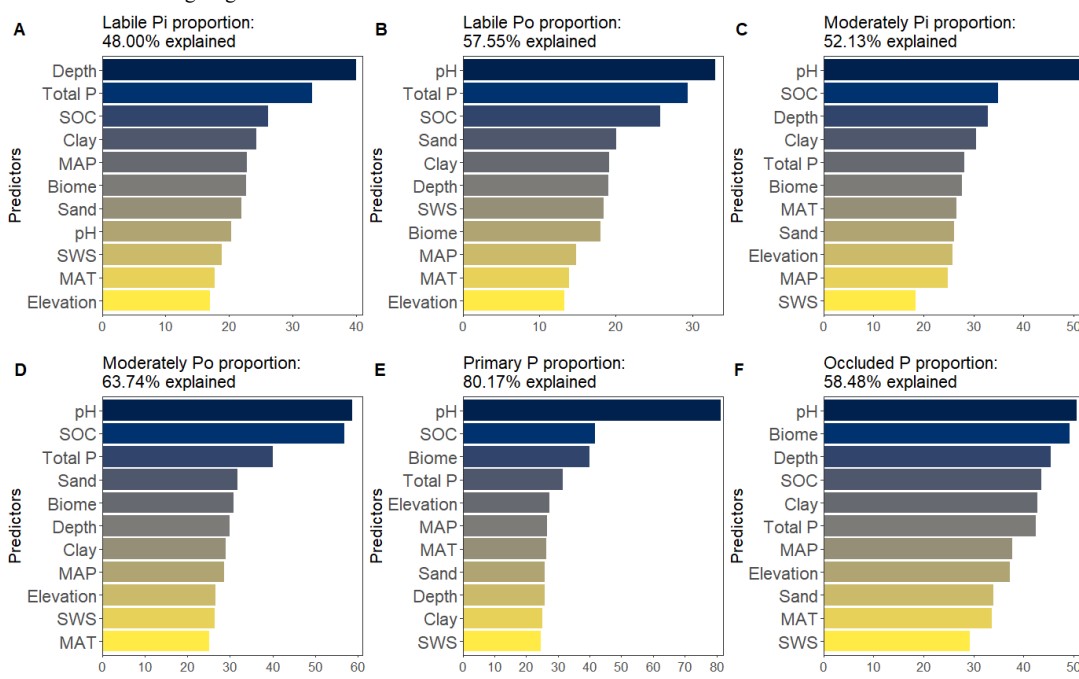





**Figure 4. Average concentrations of P pools and their proportions of total soil P concentration across soil weathering**
**stages and biomes. Labile and moderately labile Po form the organic pool.** Results based on global estimates for 0-30 cm
depth. Dry vegetation combines grassland and savanna biomes to simplify the figure.

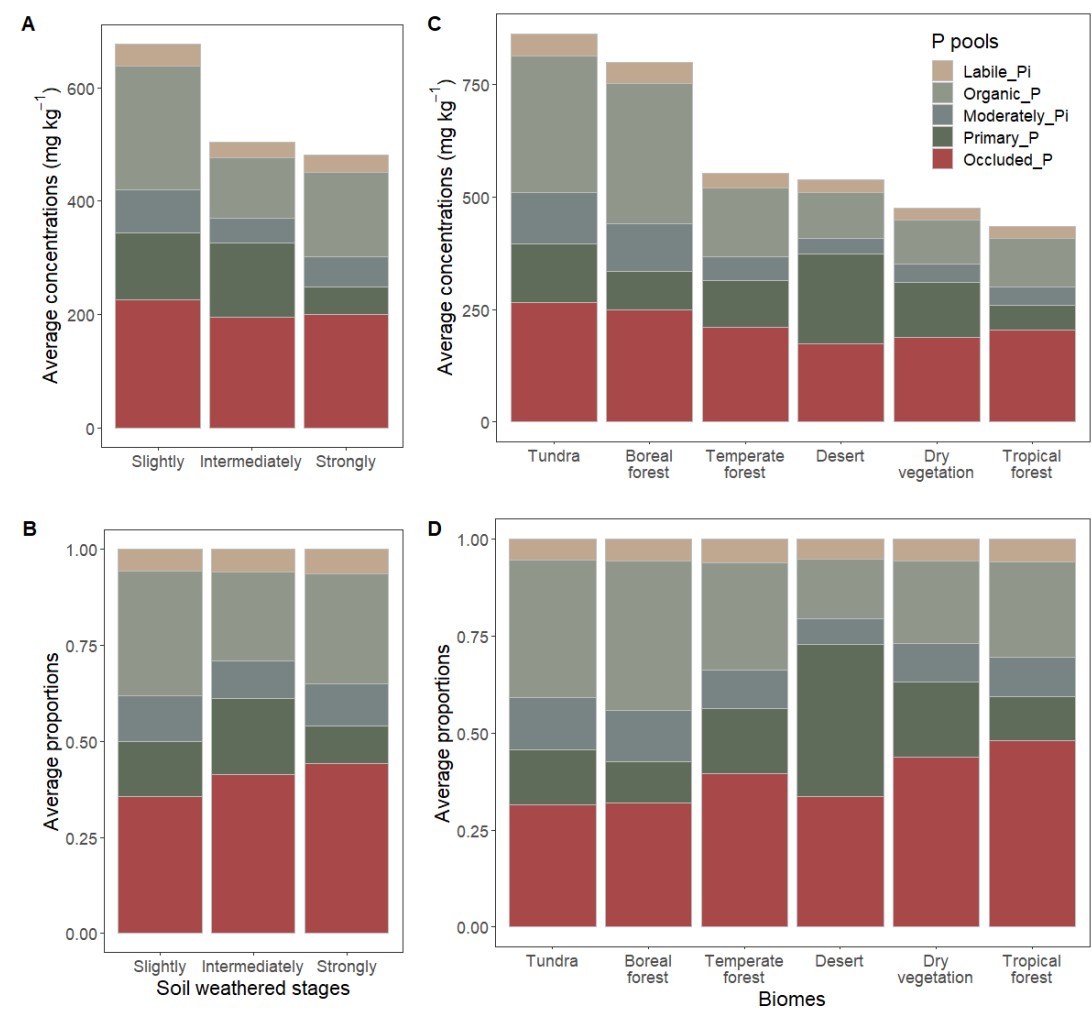








**Figure 5. Partial dependence plots showing dependence of centralized soil P pool concentrations on predictors.** To
simplify comparison, partial dependent analysis results are centralized in this plot.







**Figure 6. Partial dependence plots showing the dependence of soil P pool proportions on predictors.** The sum of all six
P pool proportions was standardized to 1, though the sum from partial dependent analysis is usually not equal to 1 due to
uncertainty.




**Figure 7. Global maps of P pool concentrations at depths of 0-30 cm.** Note that croplands and other heavily influenced areas were not masked from the maps, so soils in these areas can be used to represent soils without extensive anthropogenic activity.

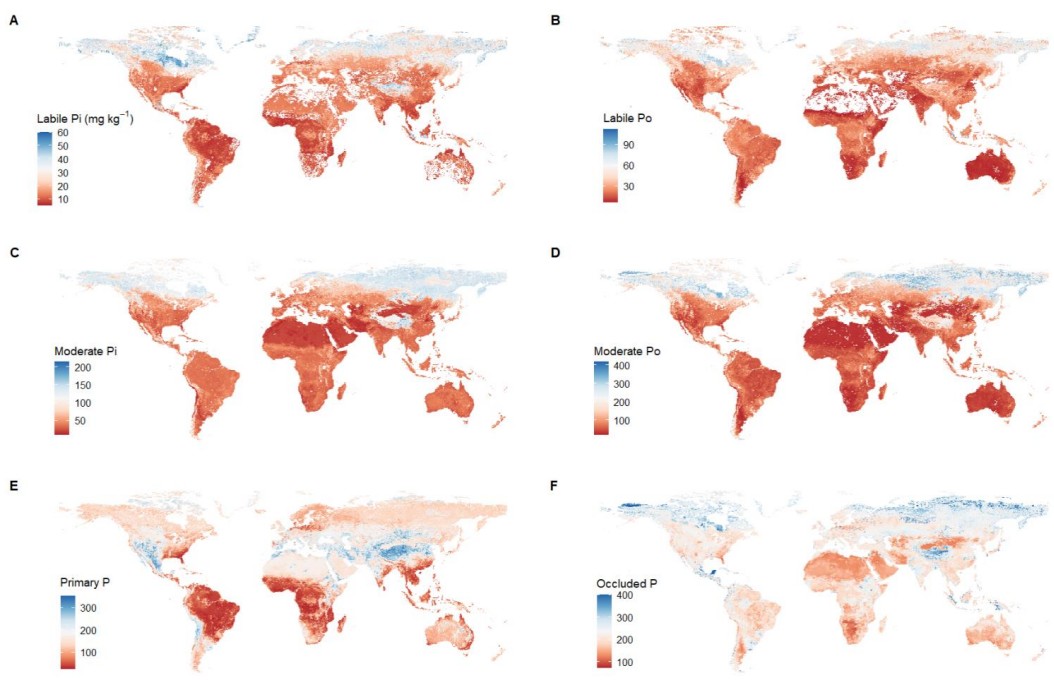



**Figure 8. Global maps of P pool proportions at depths of 0-30 cm.** Note that croplands and other heavily influenced areas
were not masked from the maps.

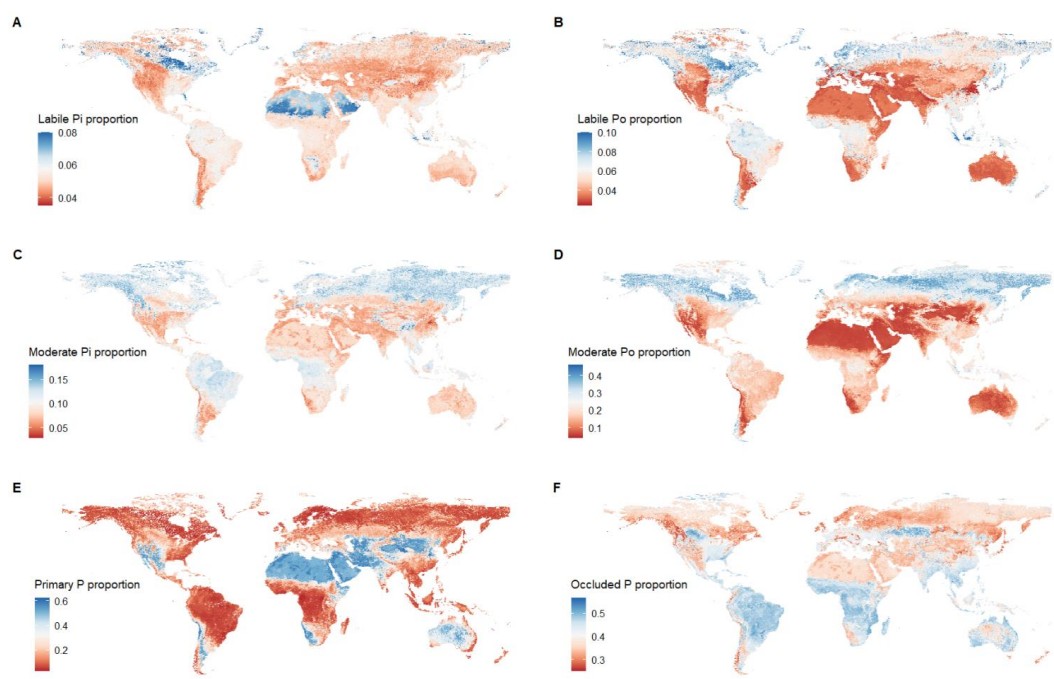

