# Peer review of "Global patterns and drivers of phosphorus fractions in natural soils 1"

_Biogeosciences, 2023_

## Author Response (AR1)

**Referee #1:**

General comments

The manuscript submitted by He et al. compiled a large database of soil P pools by the Hedley's chemical extraction method along with 11 environmental variables in global (semi-)natural ecosystems to reveal the global patterns of soil P pools and their drivers. The authors found that soil TP and pH as the main predictors for soil P pool concentrations, but soil pH and soil depth explained the variations in soil P pool proportions. Moreover, the authors presented the advantages of this database over previous one and highlighted the limitations and uncertainty of this database.

In general, this study is very interesting and important for biogeochemists to elevate the understanding of soil P cycling at the global scale. The authors did a great job to collect such a large database and take a systematic analysis of the data. Moreover, the manuscript was generally well organized with a smooth language. I think it is suitable for publication in the journal. Before recommending accepting the manuscript, I have several major concerns and some specific comments for the authors to improve the paper.

Major concerns

**Comment 1:** My main concern is that there are very different drivers for the concentrations (TP for most P fractions but soil pH for primary mineral P) vs. proportions (soil pH for most P fractions but soil depth for labile Pi) of soil P pools. Meanwhile, these factors were separately discussed in the paper (i.e., in Lines 322-341). This is beyond the common thoughts that the proportion of each P fraction is closely related to TP, because the calculation of soil P proportion in a soil or an ecosystem is based on the TP. More importantly, the authors even found the opposite trends of soil P concentrations to P pool proportions (Lines 241-243). Taking the pH (as the authors discussed) as example, soil pH can regulate all the processes of soil P pool concentrations (Lines 333-341), but why only proportions showed the close relationship? I think this difference is probably associated with the data extraction and calculation methods (Line 205-210). The proportions of each P pool were obtained by the predicted P pool concentrations rather than the measured data. But for the measured data (also the authors mentioned the limitations, Lines 246-247), it is clear that the numbers of P pool concentrations and proportions were not uniform (Table 2).

**Response 1:** Thank you very much for this comment. In contrast to the previous version, where we primarily focused on the significance of soil pH in predicting the proportions of soil P pools, we now acknowledge the importance of considering additional predictors. By neglecting to mention the relevance of these other factors, the referee may have inferred that the proportions of P pools in our findings were not strongly correlated with total P. However, our results (Fig. S4 in the revision) demonstrate that variables such as soil total P, SOC, and depth also play a vital role in predicting the proportion of P pools. Hence, we have rephrased our conclusions regarding the predictors of P pools' proportion to incorporate these significant factors:

*"When expressed in relative values (proportion of total P), the model showed that soil pH is generally the most important predictor for proportions of all soil P pools, with also prominent influences of soil organic carbon, total P concentration, soil depth and biome. These results suggest that, while concentration values of P pools logically strongly depend on soil total P concentration, the relative values of the different pools are modulated by other soil properties and the environmental context."*

We have updated this result in the Abstract (Line 32-36), Results (Lines 237-240), and Conclusion (Lines 402-403). All the line numbers in the response point to the line numbers in the changes tracked manuscript file.

The referee mentioned "the authors even found the opposite trends of soil P concentrations to P pool proportions". We believe it is rational and a good example to illustrate the divergent trends of concentration and proportion. Considering soil P depletion with soil development, soil total P and each P pool concentrations decreased with soil weathering stage; however, more proportion of P becomes occluded and we found the occluded P proportion increased with soil development. This result is discussed in the effect of soil development on P pools (Lines 362-368).

The correlation analysis results in the Table 3 are based on global predictions of these P pools concentrations, proportions, and their predictors (not the model training data in the Table 2). Thus, the numbers of P pool concentrations and proportions were uniform.

**Comment 2:** For the drivers, the authors highlighted the importance of edaphic properties and climatic factors, but the effects of climate on soil P pools were not discussed like other factors such as soil pH and development (Lines 296-298).

**Response 2:** Many thanks for the suggestion. In the revision, we have added one paragraph to discuss the effect of climate on the soil P pools (Lines 346-359), which is attached below.

**Effects of climate on P pools**

Our global predictions indicated negative effects of climatic factors (i.e., MAT and MAP) on the soil P concentrations, which means a decrease in soil P concentrations as MAT increases from northern cold biomes (e.g., tundra and boreal forest) to warm tropical biome (e.g., tropical forest) or MAP increases from arid to humid regions. These results fit well with our understanding of broad P concentration variation with increasing weathering (Walker and Syers, 1976). Also, these results are expected as the main factor determining soil P pools concentrations, soil total P, shows a similar pattern (He et al., 2021). Interestingly, we found contrasting responses of labile Pi pool's proportions along the MAT and MAP gradients. The positive correlations between labile Pi proportion and both MAT and MAP indicated labile Pi concentration decreased slower than the soil total P as temperature and precipitation increasing. This result supported the idea that biological systems evolved to retain soil labile Pi levels despite overall decrease in total soil P as long as climate factors are favorable for biological activity. In strongly weathered soil with limited soil P stocks but otherwise optimal growing conditions like in warm and humid tropical forests, the mineralization of Po and mobilization of moderately labile Pi or occluded P could contribute to maintain high levels of labile Pi due to the high soil temperature for soil enzyme kinetics and abundant carbohydrate supply from photosynthesis to fueling biological activity (Vitousek, 1984; Achat et al., 2009; Chacon et al., 2006; Liptzin and Silver, 2009).

**Comment 3:** I think the main issue lies in that many statistical methods or models were used in this study, and some may give the similar (e.g., methods in Fig. 5 and Table 3) but a little difference in the results, which results in the complex explanations for each P pools or proportion. I suggest to simplifying the methods (i.e., combining the relative importance analysis with the correlation analysis, one was used to find the main relationships between variables with positive or negative correlation, and another give the relative importance) to extract the key factors.

**Response 3:** Thanks a lot for this valuable suggestion. According to this suggestion, in the revision, we simplify the analysis methods. Now, in the main text, we use random forest regression results (Fig. 3) to indicate the relative importance and correlation analysis to indicate the positive or negative relation between soil P pools and environmental factors (Table 3). The results of partial dependent plots were moved to supplementary (Fig. S3) as a supporting material.

**Comment 4:** Meanwhile, I do not think that only the first is the key factor, and the following one or two with the high explanation degree is also the key one(s). This, to some extent, will exhibit the roles of climatic factors or even plants in soil P pools.

**Response 4:** This is an excellent comment. Many thanks. We have rephrased the text to highlight the role of factors other than the highest ranked one. For example, in the abstract, we concluded that "These results suggest that, while concentration values of P pools logically strongly depend on soil total P concentration, the relative values of the different pools are modulated by other soil properties and the environmental context." (Lines 33-35); In the conclusion, "For predicting proportions of different P pools, soil pH and to a lesser extent soil depth, SOC and total P were the most important predictors for all P pools proportions at the global scale." (Lines 402-403).

**Comment 5:** Still, for the drivers, I do not find how soil depth affected the P pools in this study. First, you did not give specific data or figures/tables to show the difference in soil P pools despite of concentrations or proportions. Second, how depth determined P pools was not analyzed well like other drivers. The discussion now can be realized without this work. My suggestion is that soil depth can be discussed along with soil development, both of which change uniformly and jointly mediate the variations in soil P pools.

**Response 5:** The variation of P pools along soil depth can now be found in the partial dependence plots (Fig. S3E). The trends of P pools' variation with soil depth are the basis for choosing the soil depth for prediction. From the results, we found changes of P pools in top 50 cm soil, but not in deeper layers (50-100 cm). Thus, we predicted P pools at 0, 10, 20, 30, 50, and 100 cm respectively to capture the changing trends in top soils. In the revision, we added these descriptions in the results (Lines 262-266).

As for the discussion, we kept the soil depth effect in a separate section, i.e., a section with a sub-title of "*Effects of soil depth on P pools*". The discussion on the effect of soil development is already long enough, which including two paragraphs in current version. And in the discussion of soil depth, we mentioned a different mechanism 'biologic uplift'. It makes the text easier to follow keeping it in a separate section.

**Comment 6:** There are 26 tables and figures (11 in the main text and 15 in the supplementary materials) in the paper, which makes the readers difficult to quickly catch the story in this study. More importantly, some figures (e.g., Fig. S9 and S10) were shown, but they were not introduced in the main text. And, the sequence of some figures was even wrong (e.g., Line 97, 207, 221). In addition, the introduction of the contents in each figure should be continuously. For example, in Lines 249-252, when you described the contents in Fig. 4, the content in Fig. 4A should be first but not those in Fig. 4C. Similarly, the Figs 5, 6 were not introduced, but the Figs 7, 8 were shown first. All these make the reading very jumping and will not help readers to have a good reading.

**Response 6:** We really appreciate this comment. In the revision, we have moved most results about the P pools' proportion to the supplementary and focused mainly on the results about P pools' concentrations. We have removed figures not introduced in the main text. And the order of all figures and tables (in main text and supplementary) were checked to make them continuous in the main text.

Specific comments

**Comment 7:** Line 50: Delete "limited"

**Response 7:** Thanks. Addressed.

**Comment 8:** Lines 53-63: The advantages of the Hedley' extraction were not well introduced, which may lead to the suspicion why this work use it. Additionally, I suggest that the contents in Lines 97-101 can be put in this paragraph, which to some extent gives the better reasons for the use of the method.

**Response 8**: Many thanks. In the revision, we have added below sentences to describe the method (Lines 60-62).

*This method exploits differences in solubility to separate different 'forms' of P occurring in the soil. Though it cannot be used to identify specific discrete P compounds in the soil, this approach has proven indispensable for the study of soil P cycling and, as such, is widely used (Condron and Newman, 2011; Klotzbücher et al., 2019; Barrow et al., 2021).*

**Comment 9:** Lines 65-75: Delete the references in the brackets when you tell the authors' name and publishing year.

**Response 9**: Thanks. Addressed.

**Comment 10:** Line 80: Why did you say "only one set of global estimates"? In the last paragraph, you illustrated several global databases of P pools.

**Response 10:** Here 'one set of global estimates' indicated the global maps of P pools by Yang et al. (2013). Several databases of the P pools exist, but not maps. We clarified the text to avoid confusion in the revision (Lines 85-86).

**Comment 11:** Lines 109-110: The OH-Po can be also associated with soil organic matters or mineral-organic complex.

**Response 11:** Thanks a lot for this reminder. In the revision, we added a word 'mainly' to make the description not that absolute (Line 80-81).

**Comment 12:** Lines 119-120: Did you check the extraction efficiency of P in different soils? Although it is not the main objective in this work, I think this way may give the data quality for a sample.

**Response 12:** This is an excellent idea. But unfortunately, we didn't include the extraction efficiency of P in the database, as this information is rarely reported in source reference.

**Comment 13:** Line 154: Biome types are fine, but why not use the productivity index (e.g., NDVI)? That may be better close to soil P pools.

**Response 13:** We explored the use of NPP (which is derived from NDVI) as a predictor, but it was disregarded given low ranking of predictor importance. NDVI like NPP are closely related to MAT and MAP which were selected predictors. No changes done.

**Comment 14:** Lines 172-173: Yes, as you mentioned in Lines 374-377, I think this is an important reason why your model sometimes only explained 48%~60% of the variance (Line 296).

**Response 14:** Indeed, it probably would increase the models' predicting ability if we would include soil extractable aluminum and iron concentrations as predictors (e.g., Wang et al 2022). However, these two variables were rarely reported and there is not a global map of them. Thus, we cannot include them in present study.

Wang, Y., Huang, Y., Augusto, L., Goll, D. S., Helfenstein, J., and Hou, E.: Toward a Global Model for Soil Inorganic Phosphorus Dynamics: Dependence of Exchange Kinetics and Soil Bioavailability on Soil Physicochemical Properties, Global

Biogeochemical Cycles, 36, https://doi.org/10.1029/2021GB007061, 2022.

**Comment 15:** Lines 178 &180: In Table 2, the largest P concentration shocked me. I

think you are right not to consider it in your model. But, you should show the data between 1% and 99% in the table.

**Response 15**: Thanks for the suggestion. In the revision, we have shown the data between 1% and 99% in the table 2.

**Comment 16:** Line 214: For the soil depth, how to understand the 0 cm? In Line 220, how is the 450 cm from? You only gave the range of 0-100 cm (see Lines 213-215).

**Response 16**: We generated predictions at five standard depths for all soil P pool concentration: 0 cm, 10 cm, 20 cm, 30cm, 50 cm, and 100 cm, which is easy for user to derive average values in a depth interval (e.g., 0-30 cm or 0-100 cm) by calculating the weighted average of the predictions within the interval. This method is used in the widely used soil gridded data *SoilGrids250m* (Hengl et al., 2017). Here, the prediction at 0 cm means setting the predictor soil depth as 0 cm.

In the result we described the database we compiled including observations from the

0.5 cm to a depth of 450 cm (Lines 224; Table 1). The training of the model was done using observations in 0 - 100 cm. This is now clarified in the text to avoid confusion (Lines 184-186).

Hengl, T., Mendes, D.J.J., Heuvelink, G.B., Ruiperez, G.M., Kilibarda, M., Blagotic,

A., Shangguan, W., Wright, M.N., Geng, X., Bauer-Marschallinger, B., Guevara,

M.A., Vargas, R., MacMillan, R.A., Batjes, N.H., Leenaars, J.G., Ribeiro, E.,

Wheeler, I., Mantel, S. and Kempen, B.: SoilGrids250m: Global gridded soil information based on machine learning. Plos One, 12, e0169748, https://doi.org/10.1371/journal.pone.0169748, 2017.

**Comment 17:** Lines 221-227: Do not repeat the data (which has shown in the table)

in the results, and just show their characteristics.

**Response 17**: Thanks. We re-wrote this paragraph (Lines 226-229).

**Comment 18:** Line 238: The sub-title is not closely related to the contents as following (introduce the drivers of soil P pools). Maybe, it is better to only highlight the drivers of soil P pools.

**Response 18**: Many thanks for this excellent suggestion. In the revision, we added drivers in the sub-title (Line 240). We kept the pattern in the sub-title, as the variation of soil P pools among different soil weathering stages and biomes is also a description of patterns of these soil P pools and this is also corresponding to the paper main title. The subtitle now reads: *3.3 Global patterns and drivers of P pools in natural soils*.

**Comment 19:** Lines 241-248 & 253-258: I suggest not to discussing the data here, and only show the main results or findings.

**Response 19**: Thanks. In the revision, we refined the text. Now we focus on the results rather than discussion. The discussion is now on Lines 294-297.

**Comment 20:** Lines 159 & 267: I do not know the difference in these two methods simultaneously using here. As I see, the correlation analysis tells us more information than that of partial dependence analysis.

**Response 20**: The main difference between partial dependence plots and linear regression is that partial dependence plots visualize the effect of an input variable on the target variable while accounting for the effects of other input variables, while linear regression models the relationship between a dependent variable and one or more independent variables, assuming a linear relationship. The partial dependence plots show non-linear relationships which cannot be resolved by the linear regression, e.g., saturation of most P concentration with increasing SOC (Fig S5b). To simplify, in the revision, we moved the partial dependence plots to the supplementary.

**Comment 21:** Line 313: What results support this conclusion?

**Response 21**: We really appreciate you pointing this out. It was a mistake. We have removed this sentence in the revision.

**Referee #2:**

General comments:

**Comment 1:**

The manuscript describes an effort to update and improve our knowledge of the global distribution of soil phosphorus much of which may be in forms not readily accessible to plants. The text is well and concisely written. More detail would be helpful, however, for aspects of the statistical analysis, interpolation from training sites to global maps and the assertion that this analysis brings us closer to the 'true' distribution patterns. By contrast, the number of tables and figures could be reduced to help the reader focus on key findings.

**Response 1**: We thank the reviewer for the positive evaluation. In the revision, we have added sentences to explain our method according referee's detailed comments below. And we have removed two figures and moved two figures to the SI from main text, and removed two SI-tables and four SI-Figures. We believe the manuscript is more concise now.

Specific comments:

**Comment 2:** I see the extraction flow chart in supplementary (Fig S3), but I would advocate promoting that (or a simplified version) to the main text. That would aid the non-specialist in thinking about the pools and how easily extractable they might be. For example at L356, the reader has to think what a 1M HCl P pool might be.

**Response 2**: Thanks for the suggestion. In the revision, we moved the extraction flow chart to the main text (Fig. 1 in the revision). And we also revised the '1 M HCl P pool' in the line 356 in previous version to 'primary mineral P' now in the revision to make it easier to understand (Line 373).

All the line numbers in the response point to the line numbers in the changes tracked manuscript file.

**Comment 3:** The reader's knowledge of random forest models is assumed, but I think more help is needed for the non-specialist. Why does a forest model tell us about soils? What is the meaning of 'tree' at L185? Also, for partial dependence plots – the idea (like partial residuals?) is to visualize the effect of a particular variable after

'controlling' for all other components of the model? By holding them at median value?

**Response 3**: In the revision, we give more detailed explanation to the random forest regression (Lines 184-186) and partial dependence plot (Lines 197-199), which are attached below. We believe they are easier to follow now.

*We used random forest regression models (Breiman, 2001) to predict global patterns of distribution for individual soil P pools. It is a type of ensemble learning algorithm that combines multiple decision trees to make predictions. It reduces the risk of overfitting and improves the generalization performance by using random subsets of input variables and training data. The output is the average prediction of all the trees (James et al., 2013).*

*Partial dependence plots are a graphical technique used in machine learning to show how the value of a particular input variable affects the predictions of a model, while holding all other input variables constant at their average values in the training data (James et al., 2013).*

**Comment 4:** The text of the results section is admirably brief, but there were passages that seemed merely to repeat detail available in the tables (e.g. L224-227; L259-266). I would encourage you to try rewriting here to provide a narrative – a story for the reader to follow.

**Response 4**: Many thanks. In the revision, we have re-organized the figures and re-wrote the results you mentioned (Lines 226-229). Now, the results are easier to follow.

**Comment 5:** Repeated references are made to the earlier study by Yang et al. (2013) and claims made that the analysis presented here offers an improvement in accuracy (e.g. L277, L284-286), but we have to go to the supplementary files to find any detail. I would suggest that some of that detail (e.g. Fig S9) be promoted to the main text. How do we know that these predictions offer 'significant improvements over earlier estimates' – where is that demonstrated?

**Response 5**: Many thanks for this excellent suggestion. In the revision, we move the Fig. S9 in the previous version to the main text (Fig. 6 in the current version), which indicates the correlation between our predictions and Yang et al. (2013).

**Comment 6:** There were too many figures for me – all six pools as concentrations and proportions. I understand that you are interested to consider absolute and relative pools, but could the presentations be simplified (with other subplots relegated to supplementary)? I had great difficulty understanding Figure 5.

**Response 6**: Indeed, there were too many figures and tables in the previous manuscript. In the revision, we have moved all results about the P pools' proportion to the supplementary and focused mainly on the results about P pools' concentrations, which makes the story in the main text easier to follow. As the results from these partial dependence plots and correlation analysis are generally consistent. To simply our results and make it easier to follow, in the revision, we moved the partial dependence plots to the supplementary. We also removed two tables and four figures from the SI, which are not introduced or not important. In current version, we have three tables and six figures in the main text; and two tables and six figures in the SI. We believe the manuscript is more concise now.

Figure 5 (the partial dependence plots) in the previous manuscript, which is moved to the SI as Fig. S3, can reveal the non-linear relations between P pool and predictor, while the correlation analysis only shows linear relation between them. Results from these two methods can corroborate each other. In the revision, we have added sentences to explain the partial dependence plot method (Lines 197-199).

**Comment 7:** The jump from Figure 1 to Figure 7 was not clear to me. The whole process of interpolation seems to be covered by an R package (L198-200). What is this, how does it work?

**Response 7**: We re-wrote the sentences describing how to use trained random forest models to generate the global predictions of soil P pools as below (Lines 202-204 in the manuscript) to make it easier to understand.

*Finally, we applied the above trained models for each of the soil P pools to global databases of the 11 predictors to generate global predictions of each soil P pools. The gridded predictors variables used for the global prediction were all re-gridded to a spatial resolution of 0.5◦ × 0.5◦ (the original resolution can be found in Table S1).*

**Comment 8:** Technical points: L25: 'random forest regression models' requires some explanation for the non-specialist. Include a clause to briefly explain what the technique attempts (see also L91, L93).

**Response 8**: Thanks. In the revision, we have added sentences (Lines 27-30) to explain the random forest model as below.

*In order to quantify the relative importance of 11 soil-forming variables in predicting soil P pools concentrations and then make further predictions at the global scale, we trained random forest regression models for each of the P pools and captured observed variation with R2 higher than 60%.*

**Comment 9:** L50: why do we only learn about pools from chronosequences?

**Response 9**: This sentence is re-phrased. We meant our knowledge about soil P pools was set up from studies on the chronosequences.

**Comment 10:** L56-58: these extraction details are rather obscure – e.g. resin versus sodium hydroxide. I think you need a line or two to outline a stepped protocol that proceeds from easy extractable pools onwards to the need for more drastic measures.

**Response 10**: Indeed, these description about different soil P pools appeared abruptly and obscure here. In the revision, we added one sentence to outline the method exploits differences in solubility to separate different 'forms' of P occurring in the soil (Line 60). In addition, we removed those specific P pools definition, which will be described in detail in the method section (section 2.1).

**Comment 11:** L75: here I think you need to include an explanation (from Hou et al.) as to the proposed mechanisms underlying these observed relationships.

**Response 11**: This is a great suggestion. Many thanks. In the revision, we added an explanation to the pattern (Lines 80-81), which is attached below.

*Hou et al. (2018a) used a global dataset compiled from analyses of 802 soil samples to examine climate effects on the soil P cycle and P availability and found that soil labile Pi concentration decreased with increasing mean annual temperature, which was mainly due to decreasing soil organic P and primary mineral P with increasing temperature.*

**Comment 12:** L104-112: have another look at your punctuation scheme here – the use of stops versus semi-colons struck me as inconsistent.

**Response 12**: Many thanks for this comment. We have revised the punctuation scheme in this paragraph (Lines 105-113). Now, the stops are used to separate different P pools. And the semi-colons are used to separate closely related independent clauses that describe the same P pool.

**Comment 13:** L113: does this paragraph not simply repeat what precedes?

**Response 13**: In this paragraph, we described we defined six soil P pools in the present study. As actually there are more than six pools in some source reference, as we described in previous paragraph, here we need this paragraph (Lines 114-118) to tell the reader how we deal with various pools in the source data. In the revision, we move the extraction flow chart to the main text (Fig. 1), from which we can distinguish the different meanings of these two paragraphs.

**Comment 14:** L205: I couldn't follow this paragraph, can you try rephrasing?

**Response 14**: In this paragraph, we explained how we get the global predictions of soil P pools' proportions. As all the three referees mentioned that we had too many tables and figures, which made it hard to generate a story easy to follow. In the revision, we mainly focus on the global predictions of P pools concentrations and moved the proportions' maps to the supplementary. Thus, to avoid a misunderstanding or misleading, we removed this paragraph in the revision.

**Comment 15:** L239: Fig. 4 is not a map.

**Response 15**: Thanks for pointing this out. We revised it to "our global predictions".

**Comment 16:** L243: we come to Figures 7 & 8 before we have been told about 5 &

6.

**Response 16**: Thanks. We have re-organized the order of all figures, which are continuous now.

**Comment 17:** L249: I would start section 3.3 with this paragraph.

**Response 17**: Thanks for the good suggestion. We have moved this paragraph to the beginning of the section 3.3.

**Comment 18:** L255: I think you mean Fig S9?

**Response 18**: Many thanks for pointing this mistake out. Yes, it should be Fig. S9

here, which was addressed in line 274 in the revision.

**Comment 19:** L278: But Table 1 indicates that Total P came from the literature?

**Response 19**: Table 1 indicates that soil total P data for model training are from literature. But when producing a global prediction using the trained model, we need a global map of soil total P. So, here we meant using an improved global map of soil total P.

**Comment 20:** L288-289: where is that shown?

**Response 20**: This result was shown in Table S5. Now we indicated this clearly in the revision (Line 274).

**Comment 21:** L313-314: that seems to contradict Fig. 2A?

**Response 21**: Appreciate. It was a mistake. We have removed this sentence in the revision.

**Comment 22:** L333-335: but inhibition of soil biota and phosphatase enzymes would reduce plant available pools.

**Response 22**: Many thanks for the comment. Indeed, low soil pH in the acid soil will inhibit soil biotic and phosphatase enzymes activity, which would reduce the soil P availability. We added this mechanism to discuss the effect of soil pH on labile Pi concentration (Lines 343-345), attached below.

*Fourth, increasing soil pH is associated with enhanced adsorption of dissolved Pi to Ca and Mg, reducing the amount of labile Pi available for plants and soil microorganisms (Fink et al., 2016; Gerke, 2015). This could explain the negative relationship between soil pH and the labile Pi proportion as identified in this study. But increasing soil pH in acidic soils favors soil microbial growth and phosphatase enzymes activity, which could increase P availability. These conflicting mechanisms may be responsible to the relative low importance in predicting the spatial variation of labile Pi proportion.*

**Comment 23:** L358-360: I couldn't follow this point about tectonic uplift – you seem to argue that there is evidence of this for all weathering stages?

**Response 23**: Appreciate this comment. We realized that it could cause controversy using tectonic uplift to explain our results that labile Pi and moderately labile Pi (non-occluded P in Walker and Syers' model) formed significant proportions of total P throughout all soil orders across weathering stages. Therefore, in the revision, we removed this mechanism. As an alternative, we discussed it could mainly due to the coarse classification of weathering stages in our study; and we also mentioned the effect of dust deposition (Lines 374-378). We believe these explanations are reasonable now.

**Comment 24:** I was surprised that no reference was made to the work led by Beto Quesada in the Amazon e.g. Biogeosciences 2010 Vol. 7 Issue 5 Pages 1515-1541.

**Response 24**: We added Quesada et al. (2010) to support the discussion about increasing trend of occluded P proportion with soil development, as Quesada et al. also discussed the similar pattern in their study. By the way, we had collected Hedley P pools data from Quesada et al. (2010), which is listed in the data source reference list.

Quesada, C. A., Lloyd, J., Schwarz, M., Patiño, S., Baker, T. R., Czimczik, C., Fyllas,
  N. M., Martinelli, L., Nardoto, G. B., Schmerler, J., Santos, A. J. B., Hodnett, M.
  G., Herrera, R., Luizão, F. J., Arneth, A., Lloyd, G., Dezzeo, N., Hilke, I.,
  Kuhlmann, I., Raessler, M., Brand, W. A., Geilmann, H., Moraes Filho, J. O.,
  Carvalho, F. P., Araujo Filho, R. N., Chaves, J. E., Cruz Junior, O. F., Pimentel, T.
  P., and Paiva, R.: Variations in chemical and physical properties of Amazon forest
  soils in relation to their genesis, Biogeosciences, 7, 1515–1541,
  https://doi.org/10.5194/bg-7-1515-2010, 2010.

**Comment 25:** Table 1: missing footnotes for symbols * and #

**Response 25**: Thanks. We added the footnotes for symbols * and # in the Table 1.

**Comment 26:** Table 3: I don't think a combined table works (concentrations and proportions). Of course Moderate Po is going to correlate strongly with Moderate Po_ppn.

**Response 26**: We combined concentrations and proportions in one table mainly as it shows that concentrations and proportions do not always positively correlate unlike the reviewer suggested. E.g., occluded P conc. is negatively correlated with occluded P proportion.

**Comment 27:** Figure 7: consider inverting the heat maps, at the moment red is low.

**Response 27**: We used red color to represent low phosphorus concentration values in these maps as strongly weathered soils (e.g., Oxisols, Ferralsols, and Ultisols) with low P concentration are usually red in color. Thus, in the revision, we still used red color to indicate low values in the global maps of soil P pools concentrations.

**Referee #3:**

He et al. consolidated a database of soil P containing 1857 entries from globally distributed (semi-)natural soils and 11 related environmental variables, and developed a new global map of soil P concentrations using a random forest model. This is an important and interesting topic, for reasons the authors describe clearly in the introduction. However there are some concerns which need to be addressed before this manuscript can be published:

**Comment 1:** The "pool" is not appropriate, because this study mainly explored the concentration and proportion of different P, without discussing soil P storage or stocks.

**Response 1:** Thanks for mentioning this. Before submitting the manuscript, we have discussed internally intensively how to call the different soil P extracted in the Hedley and its modified methods (P forms, P fractions, P pools, P proportion among others). We agreed to use pool and proportion. The terms are clearly defined (in section 2.1: Lines 103-105). In response to the reviewer's concern, we have changed the title to a more common formulation using the word "phosphorus fractions". Now the title is: "Global patterns and drivers of phosphorus fractions in natural soils". We let the editor decide if this is appropriate or not. For the latter, we would be happy to receive suggestions.

Our rationale for choosing the terms is: There is a long history of using the word 'form' to describe different fractions in the procedures (e.g., Walker and Syers, 1976; Hedley et al., 1982). However, compared to results of spectroscopic method, used by a lot of recent studies, which can provide information about P bounded to different minerals and can be more accurate to indicate different 'forms' of P in soils, the results from fractionation method are more like operational pools but not the real forms. There are many studies used the word 'fraction' to describe different pools extracted during the sequential chemical fractionation (e.g., Gatiboni and Condron, 2021; Hou et al., 2014), but it is easy to be confused with the word 'proportion' in the present manuscript. This is why in our manuscript we mainly use the word 'pool' to describe different P extracted during the sequential chemical fractionation method.

**Comment 2:** For the method, it is suggested to promote the extraction flow chart to the main file and simplify the language.

**Response 2:** Thanks for the suggestion. In the revision, we moved the extraction flow chart from the supplementary to the main text, i.e., Figure 1.

**Comment 3:** Table 3, only 7 variables were used in the correlation analysis, while 11 variables were used in the random forest analysis. Why?

**Response 3:** As for the other four variables, biome and soil order are categorical variables that are not suitable for correlation analysis and results related these two variables are shown in Figure 5; we did not include elevation in the correlation analysis matrix as it is not well correlated with and not important in predicting soil P pools variation, which could make this matrix simpler given it is already so complex. We mentioned the reason in the revision (Line 690). We did not include the soil depth in the correlation as this analysis used average prediction in top 30 cm soils, which was clarified in the title of Table 3.

All the line numbers in the response point to the line numbers in the changes tracked manuscript file.

**Comment 4:** Line 241-242 "Estimated subsoil P pool concentrations showed similar patterns to those identified in the topsoil." This study merely showed the results of surface soil (0-30cm), but lack the results of deep soil (>30 cm).

**Response 4:** We produced maps of each soil P pool at various depths (i.e., 0, 10, 20, 30, 50, and 100 cm). In the paper we focus for the sake of readability on the top soil, a commonly measured depth and in which most biological activity is concentrated. To avoid a misunderstanding and make the story easier to follow, in the revision, we removed this sentence referee mentioned in this comment. Now in the current version, we mainly focus on results of P pools' concentration in topsoil in the main text.

**Comment 5:** Line 259-272, these descriptions mainly about the results of correlation analysis (Table 3). The results of partial dependence plots (Fig. 5 and Fig. 6) need to be further explored.

**Response 5:** Thanks for this suggestion. In the revision, we described now how the soil P pools changed with soil depth in the partial dependence plot.

**Comment 6:** Line 275-293, the first part of the discussion mainly compared with the result of Yang et al. (2013), please add more in-depth arguments in the revised manuscript.

**Response 6:** In the revision, we have added more discussion on the improvement of our mapping, e.g., lines 294-297, which are attached below.

*The above-named technical improvements have made it possible to produce more accurate maps. For example, while Yang et al.'s global predictions indicated that the highest organic P concentrations were found in the temperate zone, our maps suggest they are in boreal forest and tundra. This is more consistent with general understanding of global soil organic matter distribution (Hengl et al., 2017).*

**Comment 7:** The second part of the discussion mainly discussed the effects of soil total P, pH, soil development (SWS), and soil depth on the soil P concentration and proportion. However, according to the results of Fig. 2 and Fig. 3, the contribution of SWS to soil P concentration and proportion was relatively small.

**Response 7:** Soil development was not that important in the random forest regression as it is a categorical variable and has only three levels. In the random forest regression this usually leads to a low relative importance, i.e., having more levels would likely increase the importance of SWS. Thus, one must be cautious when interpreting the results. We believe that our results (Fig. 5) as discussed on Lines 360-378 are in support of the theory of the important role of SWS on soil P composition which has been mainly based on small scale data (Walker & Syres 1967; Crews et al., 1995; Quesada et al., 2010; Selmants and Hart, 2010). We believe our global scale scope provides new evidence and thus deserves focus.

**Comment 8:** There are too many figures and tables in this paper, and some figures and tables provided in the supplementary materials (e.g. Table S2, S3, and S4) are not used in the main file.

**Response 8:** We really appreciate this comment. In the revision, we have removed those tables and figures not introduced in the main text (i.e., Table S2, S3, and S4; Fig. S5, S6, and S7 in the previous version). We also have moved two tables and two figures to the SI from main text. The manuscript is more concise now. And the order of all figures and tables (in main text and supplementary) were checked to make them continuous in the main text.

---

## Author Response (AR2)

**Comment from Editor:**

Greetings. As you can one of the reviewer as raised a query regarding interpretation. Let me know the what you have to say on that. If required make the revision accordingly and also respond to the suggestion.

Submit you final revised manuscript.

**Response:**

We express our heartfelt gratitude for the editor's invaluable efforts throughout the review process of our paper.

Regarding the referee's query, we acknowledge that our previous response may not have adequately addressed the comment in R(5, 2). However, we want to assure the reviewer that we have provided a comprehensive discussion of the improvements made to our maps in comparison to previous versions in the main text. Therefore, we have not made any changes to the main text.

For more in-depth information, we kindly ask the reviewer to refer to our detailed response to the referee below.

**Comment from referee:**

Congratulations on the work undertaken in this revision to address the earlier comments.

One point (R2, 5) still strikes me as a problem, however. The Discussion opens with the assertion that your predictions result in significant improvements over earlier estimates. Your response to the initial comment suggests that Figure 6 provides the evidence for this - but that only demonstrates rather weak correlations between two sets of estimates.

I was expecting to see those predictions compared against independent measurements.

What I think you are claiming is that your study draws on a much larger body of data and includes more putative drivers.

**Response:**

We express our gratitude to the reviewer for their thorough assessment. Upon reflection, we acknowledge that our response in R2, 5 may have been misleading. However, we would like to clarify that in the discussion (Lines 276 – 287 of the final manuscript version), we present compelling arguments supporting why our estimate represents a significant improvement over existing ones. These reasons include: (1) the utilization of an improved soil total P map to predict soil P fractions, (2) a substantial increase in the number of observations on P fractions by an order of magnitude, and (3) the adoption of a mapping approach that effectively captures variation within soil orders, incorporating more putative drivers. The reviewer aptly summarizes our approach by stating, 'What I think you are claiming is that your study draws on a much larger body of data and includes more putative drivers.'

Given the limited number of data points available, we made the decision to utilize all available data for the random forest training and validation, employing a Five-fold cross-validation (Lines 191-192). This approach is commonly accepted when dealing with constrained data scenarios. Nevertheless, we concur that an evaluation based on large-scale gradients would be advantageous. Unfortunately, we encountered challenges in identifying an appropriate dataset for such an assessment. We remain open to exploring this avenue in future research endeavors.